# Microclimate Multivariate Analysis of Two Industrial Areas

**Angela Maria de Arruda** [1,*]**, António Lopes** [2,3,*] **and Érico Masiero** [1]

1    PPGEU—Postgraduate Program in Urban Engineering, NUPA—Center for Acoustic and Thermal Research in Buildings and Road Networks, Federal University of São Carlos, São Carlos 13565-905, Brazil; erico@ufscar.br
2    Center for Geographical Studies, IGOT—Institute of Geography and Spatial Planning, University of Lisbon, 1600-276 Lisbon, Portugal
3    Associate Laboratory TERRA, 3000-456 Coimbra, Portugal
*    Correspondence: angelaarruda@estudante.ufscar.br (A.M.d.A.); antonio.lopes@edu.ulisboa.pt (A.L.)

**Abstract:** Most of the existing studies on the increase in air temperature (AT) in industrial neighborhoods (UIs) approach the subject from the analysis of the land surface temperature (LST). Therefore, the objective of this study was to analyze, in addition to LST, the variables of air temperature, relative and specific humidity, wind speed and direction, sky view factor and the albedo of the material surfaces, and to verify which of them has a greater impact on the urban microclimate of the UIs of two cities, Sintra/PT and Uberlândia/BR. To develop this analysis, representative sections of industrial urban areas in the previously mentioned cities were selected and computational simulations were carried out with the ENVI-met software to obtain results related to the studied variables. The results of the simulations, analyzed using multivariate analysis, showed that even though the Udia UI has materials with lower albedo (−45%), lower percentages of vegetation (−20%) and lower WS (−40%) than the Sin UI, the AT inside it may be lower than in the unshaded surroundings around 1.3 °C. For Sin UI, a difference in WS of −1.9 m/s, compared to the control points, caused a peak of +1.5 °C in the industrial environment at 13 h, contrary to what happened in Udia UI.

**Keywords:** urban microclimate; multivariate analysis; computer simulation

## 1. Introduction

Climate change is a concern of the scientific community, citizens and the political class around the world [1], with the city being a central agent in these changes. The effect of urban heat islands (UHIs) is defined as the rise in the urban air temperature in the urban limit atmospheric layer compared to nearby suburban or rural environments [2], which has been highlighted as anthropogenic activities are transforming agricultural matrix societies into industrial and service societies [3]. Human activities and the physical characteristics of the environment, particularly those of the built environment, such as the high density of buildings, the concentration of building materials with a high energy potential, reduced evapotranspiration, greater heat storage, increased balance of solar radiation, reduced advection and increased anthropogenic heat, are its causes [2,4–9].

To understand what happens in the canopy layer, that is, between the soil and the average level of the roofs and the peculiarities of each site, it is interesting to identify the intrinsic characteristics of each urban cut. Stewart and Oke [10] proposed local climate zones (LCZs) for classifying each urban and rural landscape according to the properties of the elements present in each location. The main properties analyzed are the height of surface roughness, waterproof surface fraction, and thermal performance of materials, among others.

Stewart [11] explains that by adopting the LCZs system, the universal study class of UHIs is the landscape composed of the properties that influence the thermal field of the canopy (surface morphology—height and density of the object) and the soil cover (waterproof or permeable). The surface morphology has an impact on the local climate by

altering the airflow and heat transfer in the air, as soil cover changes the albedo, humidity availability and soil heating and/or cooling capacity. The LCZs are divided according to the type of buildings present and types of soil cover, ranging from densely built, large buildings with low height to cuts with heavy industries. Regarding the coverings, there is variation in the density occupied by trees, exposed soil, rock, and water [10].

There is widespread interest in comparing different regions, generally rural and urban centers, in terms of land occupation, waterproofing levels and vegetation. However, within the urbanized spaces, there are differences in air temperature and relative humidity, direct consequences of microclimate, as well as environmental and spatial variables. In general, UHI studies are devoted to the central areas of large cities, and they do not always consider diverse spatial configurations or the seasonality of land use and occupation. Industrial areas, in particular, are locations with the potential for UHI formation and consequent interference in nearby areas [12]. The choice of building materials in construction has a prominent role in improving the thermal conditions of the environment and mitigating the effects of UHI, the best choice being materials with lower absorption capacity, greater reflectivity (albedo) and greater thermal conductivity, called cooling materials [13].

In addition to the fresh building materials used in the envelope and roof of buildings, with the use of reflective roofs and walls [14] or green roofs [15], different strategies for mitigating the effects of UHIs have been addressed for their surroundings, such as the use of cold pavements with integrated vegetation, pavements with porous and more permeable concrete for sidewalks, the replacement of conventional pavement with concrete grass grid pavers in parking lots, and living fences and trees. The results of these replacements show a reduction of up to 3 °C in AT and 30 °C in LST, in addition to improving thermal comfort indices [16]. The decrease in TA associated with vegetation in urban canyons can reach 10 °C to 15 °C depending on the layout of trees and shading [17].

The high concentration of buildings, extensive areas built with horizontal buildings with metal covers, little or no vegetation, widely paved urban infrastructure, intense traffic and anthropogenic residual heat resulting from industrial activities have been recognized as contributing factors to the formation of UHIs, i.e., urban regions with higher temperatures that are concentrated in industrial regions due to the exploration and use of natural raw materials for the production of energy and marketable products whose direct impact on the natural ecosystem is observed through increased intraurban surface temperature [8,9,18–20]. The impact of these infrastructures on surrounding areas can drastically affect the urban climate and the comfort and health conditions of the surrounding area. The intensity of UHI is influenced by both the urban landscape and the type of urban development [8]. The different types of soil cover and the distance of industrial clusters have an impact on the cooling speed of the cuts, especially in relation to LST: large waterproofed extensions dominate the LST standard, unlike that recorded in water bodies [21]. Mohan et al. [12] obtained values above 2 °C for the intensity of nighttime UHI, varying the classes of land use and coverage being the mining sites, which presented the highest intensity (2.52 °C), followed by industrial districts (2.32 °C) and rural and urban settlements (2.13 °C). Singh et al. [22] concluded that there has been a constant accumulation of heat over the years in the industrial agglomeration in Jharsuguda/India, with an impact on the rise in nighttime air temperature, attributed to changes in land cover, intense industrial activities and mining processes. According to Meng et al. [9], the declining trend in the variation in LST from inside industrial centers to their outskirts proves the impact of such regions on UHIs. When compared to weather seasons, the greatest disparity between the LST of the spaces studied occurred during the hot seasons; the intensity of UHI was greater in spring and summer.

The identification of hot spots, for example, from the aforementioned studies, makes it possible to classify the type of local climate and different urban thermal patterns through the study of the microclimates of a city [23]. However, obtaining weather data with high time resolution, essential for the analysis of climate phenomena on a local scale or micro-



scale, in general, encounters logistical and financial obstacles, according to Reis et al. [24], making it difficult or sometimes rendering it impossible to study.

In this context, it is understood that many of the existing studies on UHIs in industrial regions address the topic from the analysis of the LST in functions of the type of coverage and land use. Therefore, it becomes interesting to expand such analysis by inserting other climate variables as a contribution to the studies of urban thermal comfort, especially in industrial districts. In this sense, some numerical and computational models are used for urban climate analysis in open spaces, highlighting the ENVI-met software, which provides results for properties such as temperature and relative humidity of the air, solar radiation, speed and direction of the wind and the albedo of the surfaces, among others. Micrometeorological thermal simulations help in understanding the dynamics of UHI formation and also in estimating the effectiveness of possible mitigation measures [18,25–29].

The choice of the cities of Uberlândia/BR and Sintra/PT is justified by the opportunity to study two locations with unique characteristics. The first one is a Brazilian city in the process of accelerated growth, in which it is possible to see an excessive process of verticalization of urban occupation in central areas and various areas with inadequate infrastructure for housing [26]. Add to these characteristics the identification of processes of expansion of extensive industrial areas in the periphery, which further suppress the vegetated areas of the municipality and contribute to changing the urban microclimate. Thus, it would be interesting to extend the microclimate study, broadly exploited in the Portuguese capital, Lisbon, to a municipality located in the metropolitan region, and that has expressive industrial zones, like Sintra.

In this study, the microclimate of urban clippings industrial is analyzed through the three-dimensional simulation software ENVI-met (V 5.0.1), which makes it possible to obtain the meteorological data. The simulation is performed through a three-dimensional model of the urban microclimate, by means of surface–vegetation–atmosphere interactions. The energy balance is calculated using the variables radiation, reflection and shading from buildings and vegetation, airflow, temperature, humidity, local turbulence and dissipation rate, as well as water and heat exchanges from the soil [30].

Thus, this study aimed to analyze which variables, in addition to LST, have the greatest impact on the urban microclimate of industrial districts in two cities, Sintra/PT and Uberlândia/BR, according to the type of climate and the physical composition of the urban clusters, using multivariate analysis with the data obtained from the ENVI-met microclimate model and later comparing them to the data of the variables obtained from meteorological stations positioned in open places, called control points, to identify the specificities of each microclimate.

## 2. Methods

To develop this study, representative clippings of industrial urban areas were selected in the cities of Sintra and Uberlândia, identifying percentages of vegetation, roofed and sealed areas, as well as the urban geometry, represented by SVF and the albedo of the surfaces, calculated from *Landsat 8* satellite images, obtained through the USGS Landsat data archive, accessed through special search tools and online request, namely USGS Global Visualization Viewer (GLOVIS) (available at http://glovis.usgs.gov, accessed on 15 January 2023), EarthExplorer (http://earthexplorer.usgs.gov/, accessed on 15 January 2023) and QGIS software to calculate the albedo of the surfaces and characterize the clippings.

The council of Sintra, in Portugal, has had a Municipal Director Plan since 1999, which includes agricultural and ecological reserve areas. The current plan, under review, defines industrial zones between mountainous and environmental conservation areas. Even with planning and laws on land use and occupation, studies suggest that agricultural areas contained in the Lisbon metropolitan area tend to be pressured to become urban, which imposes greater challenges to the urban confinement plan [31].

The metropolitan area of Lisbon comprises 18 municipalities, grouped into two sub-regions: the Grande Lisboa and the Setúbal Peninsula. The council of Sintra, whose

location is shown in Figure 1, has 319.23 km$^2$ and 385,954 inhabitants [32] and is subdivided into 11 parishes. It is the second most populous county in Portugal and presents great heterogeneity in its territory, with the coastal and northern parishes having forest and rural characteristics, while in the south there are the urbanized parishes, the object of this study. It has a temperate climate with rainy winter, dry summer and low heat (Csb), according to the Köppen–Geiger classification, with an average altitude of 206 m and an annual average temperature of approximately 15 °C [33]. (Figure 1).

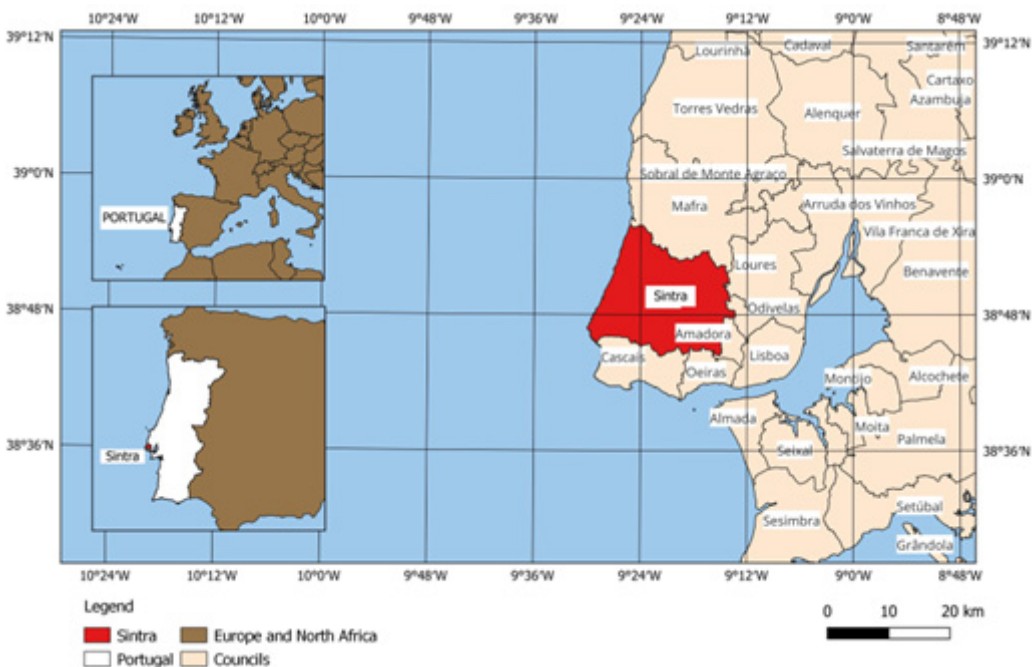

**Figure 1.** Location of Sintra/PT. Source: Lisboa Aberta (2023).

The city of Uberlândia, Brazil, is located in the interior of the state of Minas Gerais, 18°55′08″ S, 48°16′37″ O (Figure 2), with an estimated population of 706,597 inhabitants [34], and it has a total area of 4,115.206 km$^2$. It is a major regional industrial and logistical center and its economy is based on industrial and service activities. The climate is tropical with a dry season (Aw) in the Köppen–Geiger classification, an average annual temperature of 22.3 °C, an altitude of 863 m and an IDH of 0.789 [34].

The choice of the scenes, shown in Figure 3, is justified by their belonging to an industrial LCZ, called LCZ 8, similar to the two cities addressed, according to the classification principles of Stewart and Oke [10] and Demuzere et al. [35].

In Figure 4, we have the distribution of LCZs along the industrial territory and surroundings of Sintra (Sin) and Uberlândia (Udia), with LCZ 8 represented in gray, predominantly occupied by large constructions, soil covering mostly paved areas, surface albedo between 0.15 and 0.25 and the following building materials present: steel, concrete, metal and stone [10]. The residential LCZ 3 is represented by the colors red and orange, indicating high construction density with buildings of up to 2 floors and low construction density characterized by more widely spaced constructions, respectively.

Subsequently, values of the input parameters for the computer simulations were collected from Climate.OneBuilding.Org [36], such as the air temperature (AT), land surface temperature (LST), relative humidity (RH), specific humidity (SH), wind speed (WS) and wind direction (WD) of the regions in which the cities are located. Starting from the physical characteristics of the urban clippings representative of industrial areas and the climatic information, models of the urban clippings were prepared for the development of the simulations in ENVI-met software in order to identify a typical day for each area,

corresponding to the 5th percentile of the hottest days of the year, with calm, rain-free winds and little cloudiness.

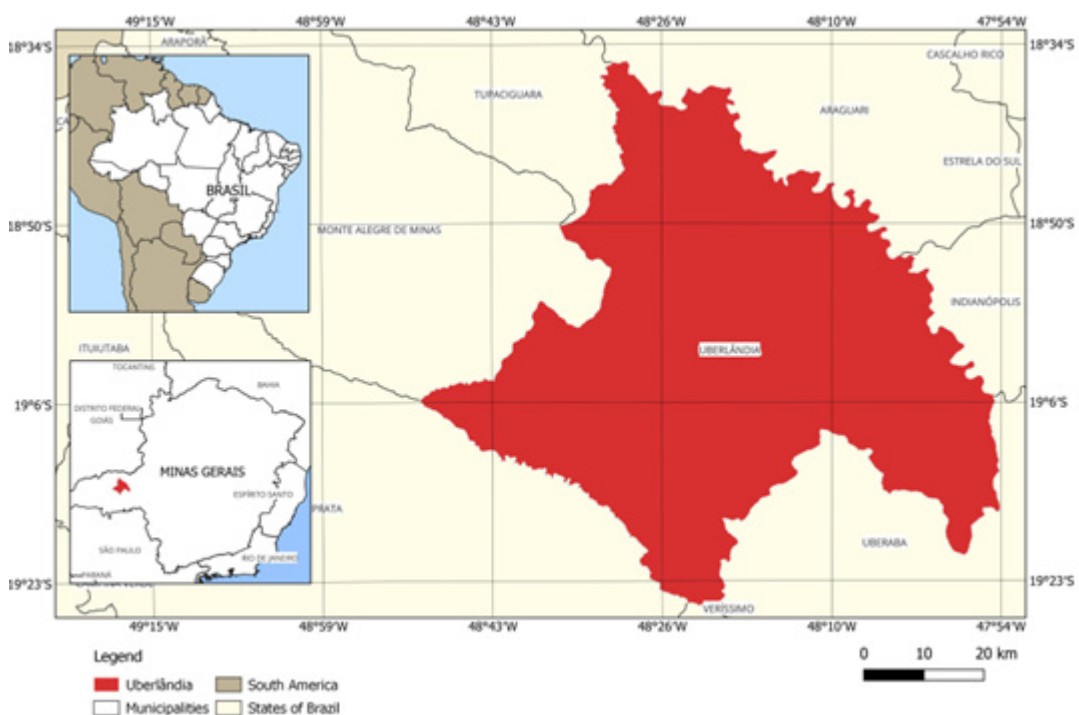

**Figure 2.** Location of Uberlândia/MG. Source: IBGE (2023).

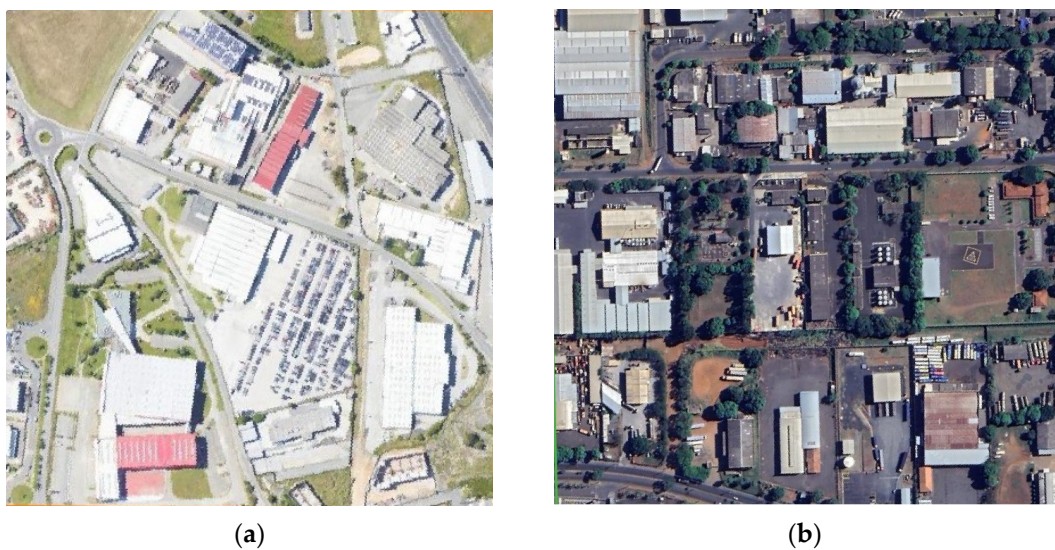

**Figure 3.** Scenes studied: (**a**) UI Sintra; (**b**) UI Udia. Source: Google Earth (2023).

It should be noted that the ENVI-met microclimatic model presents, according to Lopes et al. [37], the limitation of not yet being a viable solution for predicting the intensity of future UHIs, since it is restricted to small urban areas, around 3 km$^2$, mainly due to the computational processing capacity. In this case, the use of statistical techniques and the development of predictive models for certain climatic contexts prove interesting to spatially extend urban microclimatic studies [38]. Croce et al. [20] add that the results of microclimatic models strongly depend on the boundary conditions of the urban area under study defined in the configuration file, which may affect the simulation results.

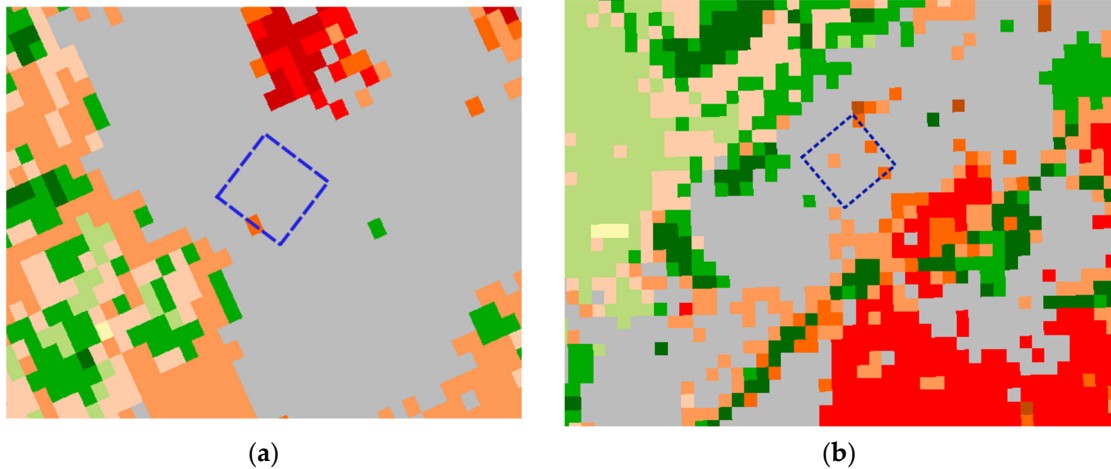

**Figure 4.** Classification of the LCZs of and Uberlândia UI Scale: 1:25,000 (**a**) UI Sintra; (**b**) UI Udia. Source: Adapted from Demuzere [35].

Figure 5 shows the summarized steps of the method.

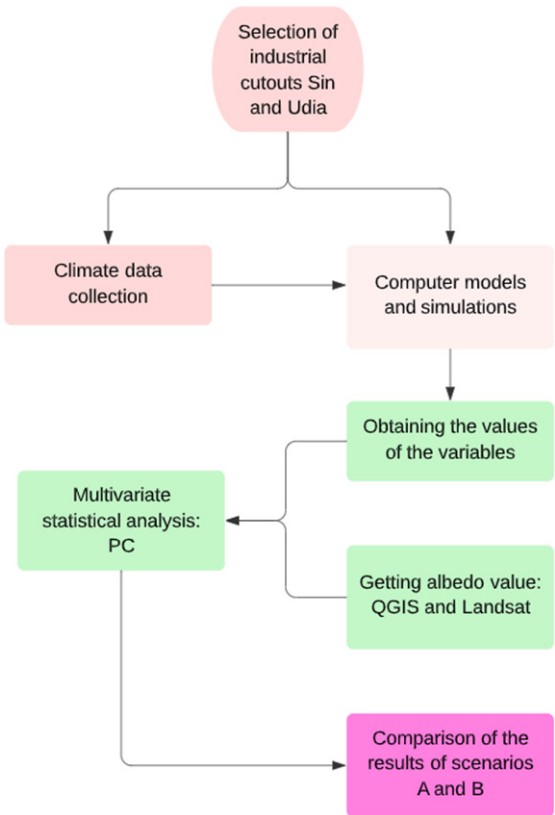

**Figure 5.** Summary of the research method.

## 2.1. Input Parameters for ENVI-Met

The values of the input parameters for the simulations were gathered from Climate.OneBuilding.Org [36], which provides the averages of specific air temperature and humidity, wind speed and direction, precipitation and radiation from the years 2007 to 2021. Such a file is inserted into the Full Forcing option in ENVI-met, and the summary input data are presented in Table 1 for the selected days.

**Table 1.** Input data.

| Category | Input | |
|---|---|---|
| Modeling area (L, W, H) (m) | 500 × 500 × 50 | |
| Grid cell (x, y, z) | 4 × 4 × 2 | |
| Cities | Uberlândia | Sintra |
| Configuration file<br>Simulation start date<br>Simulation end date<br>Simulation period | <br>05:00 h (23 January 2022)<br>04:59 h (24 January 2022)<br>24 h | <br>05:00 h (17 July 2022)<br>04:59 h (17 July 2022)<br>24 h |
| Meteorological input | AT max = 41 °C—17:00 h<br>AT min = 29 °C—06:00 h<br>SH max = 12.5 g/kg—22:00 h<br>SH min = 8 g/kg—13:00 h<br>WS max = 3.6 m/s—11:00 h<br>WS min = 0—5:00 h | AT max = 29 °C—12:00 h<br>AT min = 15 °C—05:00 h<br>SH max = 13 g/kg—13:00 h<br>SH min = 9 g/kg—21:00 h<br>WS max = 6 m/s—15:00 h<br>WS min = 0—7:00 h |
| Material | Roof—sandwich roofing sheet<br>Pavement—dark asphalt<br>Vegetation—grass and trees | Roof—sandwich roofing sheet<br>Pavement—light asphalt<br>Vegetation—grass and trees |

The choice of 23 January 2022 and 17 July 2022 for the simulations in Uberlândia and Sintra, respectively, is justified by the fact that these two days correspond to the 5th percentile of the hottest days in the past ten years with temperatures above 30 °C.

## 2.2. Albedo Calculation

To calculate the exoatmospheric albedo, we followed the method suggested by Lopes [39]. First, it is necessary to define the area, the period and the percentage of cloud cover in the database USGS Earth Explorer [40]. After selecting the Landsat 8–9 OLI/TIRS C2 L2 dataset, it is possible to select the desired images and the bands B2, B3, B4, B5 and B6 in TIF format. In the QGIS software, the algorithm of Olmedo [41] is applied through the raster calculator tool with the coefficients presented in Equation (1) to obtain an image with the albedo data calculated for each pixel:

$$\text{Albedo} = rs,B\ (0.246) + rs,G\ (0.146) + rs,R\ (0.191) + rs,NIR\ (0.304) + rs,SWIR1\ (0.105) + rs,SWIR2\ (0.008) \qquad (1)$$

where rs B is the SWIR, Landsat bands 2 to 6 (after being transformed into reflectances).

Applying this methodology, we obtain Figure 6, which shows the variation in the albedo in the Sintra and Uberlândia UIs.

## 2.3. Multivariate Analysis

After the simulations, 30 random points were extracted from each studied clipping, for the time of 2 p.m., at a height of 9 m from the ground (Figure 7). These points were identified according to the initials of the city and the location of the point, with U1 referring to point 1 of Udia and S1 referring to point 1 of Sintra, for example.

The data were organized and subsequently treated in R-Studio software. The MultivariateAnalysis package [42] was used to develop the statistical tools known as principal components and dendrograms.

For the analysis of the results, it was decided to apply multivariate analysis techniques, especially factor analysis and cluster analysis, which, according to Hongyu et al. [43], are viable and efficient alternatives for the analysis of results, making it possible to reduce the dimensionality of variables, group individuals according to their similarities, and, therefore, analyze the most important variables for each group involved. Moreover, factor analysis and cluster analysis are widely used in studies of climatological phenomena and identification of climate zones, such as Zscheischer et al. [44], Amiri, Mesgari [45], Silva et al. [46], Leoni et al. [47], Nunes and Sousa [48], Praene et al. [49] and Valverde et al. [50].

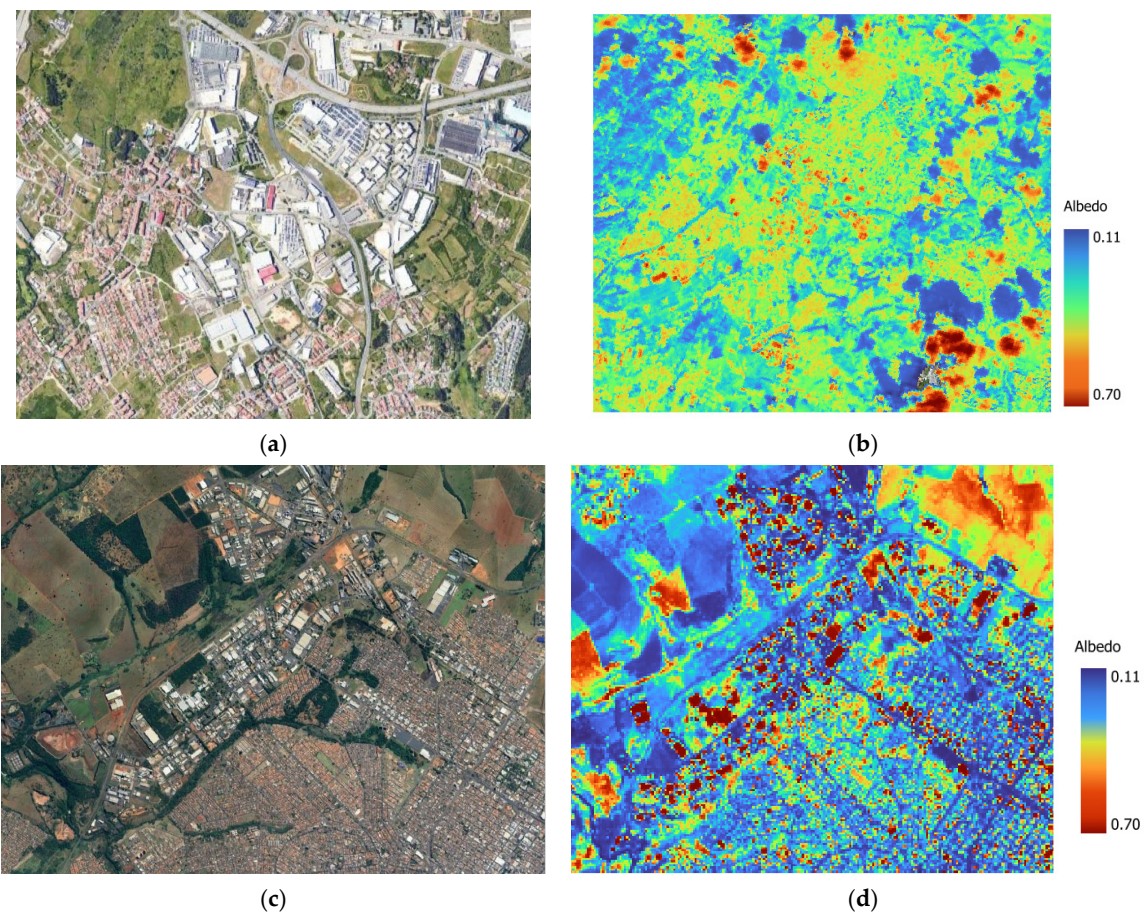

**Figure 6.** Sintra and Udia UIs and albedo: (**a**) UI Sintra; (**b**) Landsat 8 image of Sintra; (**c**) UI Udia; (**d**) Landsat 8 image of Udia.

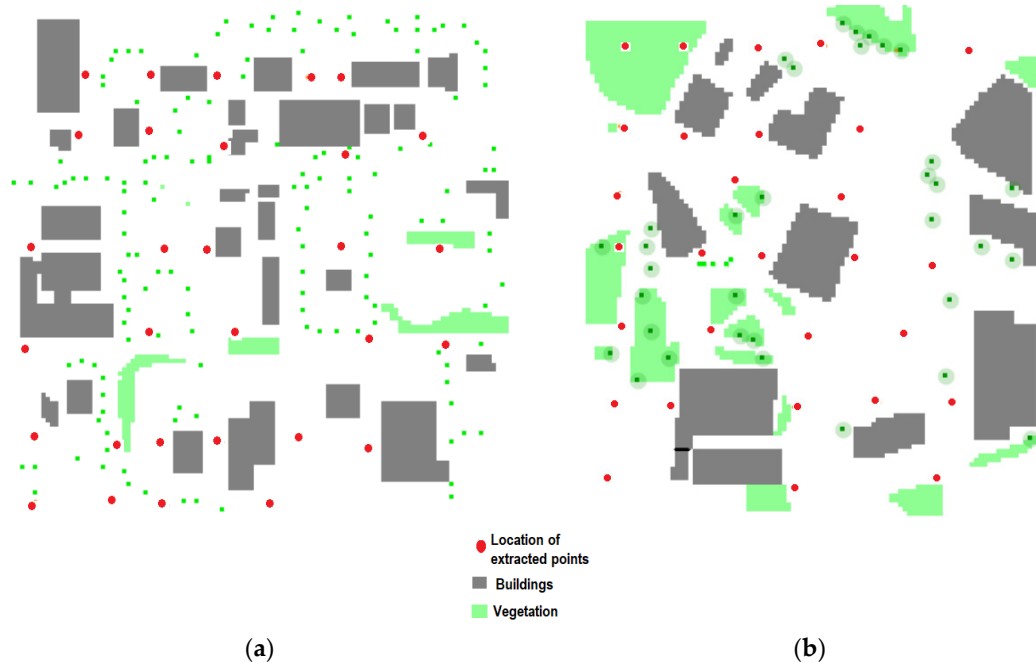

**Figure 7.** Locations where the study points were extracted: (**a**) Udia; (**b**) Sin.

Principal component analysis (PCA) is one of the most widespread methods of factor analysis. It uses the modeling of the covariance matrix of the data, and it consists of linearly transforming a set of original variables, correlated among themselves, into linear combinations of them, linearly independent, reducing the total data and extracting only those that describe most of the total variability, with as little data loss as possible [43].

For the choice of principal components (PCs) to be retained, one can resort to the screen plot, where one searches the graph that represents the eigenvalue versus the percentage of variance explained, the "jumping off point" from which the PCs do not have so much importance for the total variance [46]; the Kaiser criterion [51], which selects the PCs according to the value of their eigenvalues, with eigenvalues > 1 belonging to the PCs that explain the greatest variation in the total sample space [43]; or the analysis of representativeness in relation to the total variance, in which the researcher predetermines a percentage of explanation of the total variance, according to the phenomenon studied, with commonly accepted values being above 70%.

Associated with PCA, cluster analysis is used to divide the sample elements into groups according to their proximity and common characteristics. The distance between two objects can be measured according to several criteria, with the Euclidean distance being the most common and appropriate dissimilarity measure for quantitative variables: the higher this value, the more different, or dissimilar, the objects are [48]. The hierarchical clustering method, on the other hand, groups similar elements and the process is repeated at various levels, forming a tree called a dendrogram [48]. The link between elements can be the nearest neighbor, farthest neighbor or average link.

After clustering, it is possible to evaluate the degree to which the original distances and the paired distances are maintained by the dendrogram using the cophenetic correlation coefficient, where a value closer to 1 means better correlation. The existence of similar behaviors between variables and the formation of internally homogeneous and internally heterogeneous groups from the study of certain variables can be verified through such exploratory tools [47].

## 3. Results

In this paper, the results are structured in three sections:

- Hierarchical clustering and principal component analysis (PCA): Assessment to reveal similarities in data distribution patterns and establish possible associations between the main variables in the microclimate study of each clipping and the impact of each variable for the scenario studied;
- Physical composition of the clippings and albedo: Presentation of the physical characterization and albedo of the clippings, which influence the urban microclimate;
- Descriptive analysis: Assessment of the simulation results for the variables with the greatest impact on each scenario.

### 3.1. Hierarchical Clustering and Principal Component Analysis

The results of the hierarchical clustering revealed similarities in the data distribution patterns between 60 points collected at the UIs in Uberlândia and Sintra. The average method was used to determine the optimal clusters (k) and interpret and validate the data clusters. This method led to a correlation between the cophenetic distance and the original distance of 0.94, confirming that the dissimilarity matrix is well represented in the dendrogram. As a result, two main clusters were identified based on their distinctive features in the dataset (Figure 8), with sizes of 30 composed exclusively of elements from Sintra or Udia.

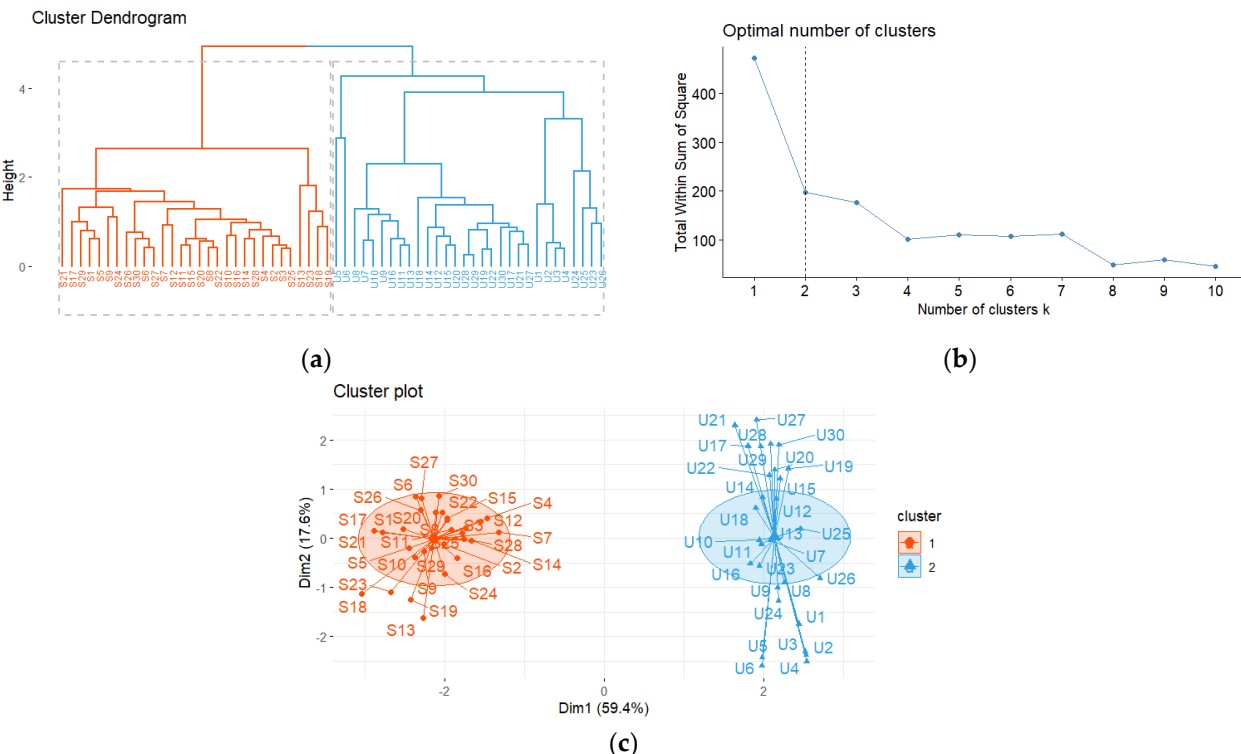

**Figure 8.** (**a**) Dendrogram showing the two clusters identified. (**b**) Screen plot showing a slow decrease in inertia after k = 2. (**c**) Visualization of the k-means cluster plot.

The research dataset initially contained eight variables: AT, RH, SH, SVF, WS, WD, albedo, LST. To simplify the descriptions for further analysis, principal component analysis (PCA) was implemented to reduce the dimensionality of the data. The variables that have the highest contribution are highlighted in the bar chart (Figure 9). The dashed line presented in the graph corresponds to the expected average percentage of the variance explained, 12.5%.

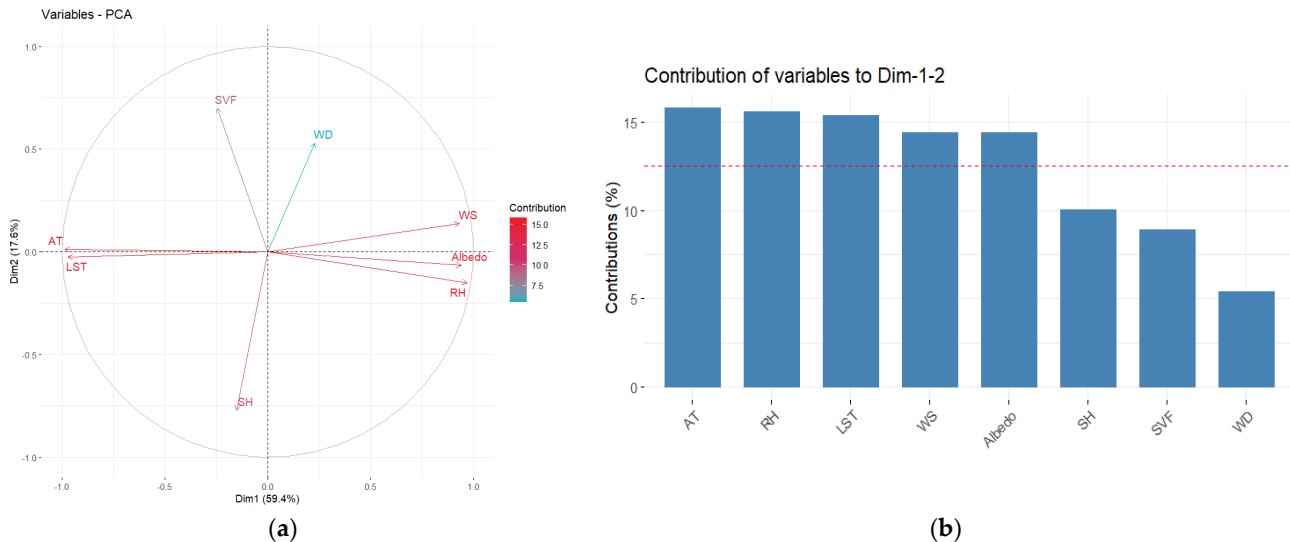

**Figure 9.** (**a**) The largest variance contribution to the PC1 and PC2 correlation circle (**a**) and bar chart of CP contribution in 8 variables (**b**).

Figure 10 shows all five PCs and the weighting coefficients of each study variable for the PC, where the first three stood out. For PC1, the variables AT, RH, WS, albedo and LST

stood out. For PC2, SH and SVF had the greatest contribution, while WD was highlighted for PC3.

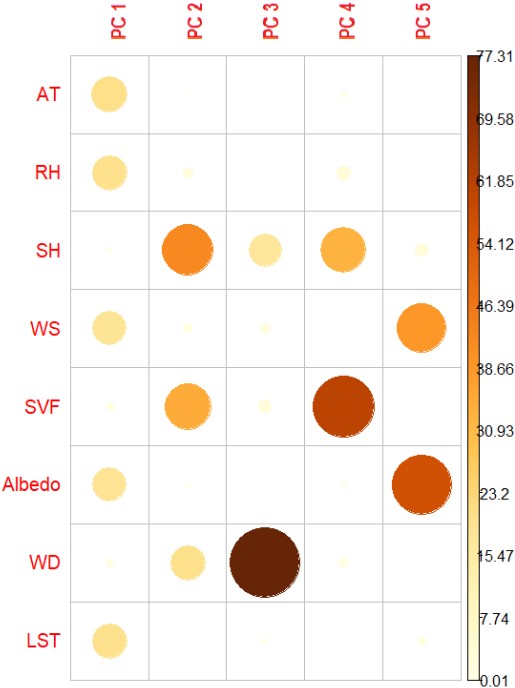

**Figure 10.** PCs and variable contributions.

As criteria for choosing the principal components, we can choose eigenvalues > 1, according to Kaiser's criterion [51], or select the components that allow explaining 70% of the variance. The first two components satisfy both criteria simultaneously, according to Table 2. The screen plot can also be used, which facilitates the visualization of the point where the explained variance tends to stabilize; that is, the first two PCs have 77.02% of the variance, and therefore, they effectively summarize the total sample variance and can be used to study the dataset (Figure 11).

**Table 2.** Eigenvalues and percent variances of principal components.

|  | Eigenvalue | Variance Percent | Cumulative Variance Percent |
|---|---|---|---|
| PC1 | 4.75 | 59.40 | 59.40 |
| PC2 | 1.40 | 17.62 | 77.02 |
| PC3 | 0.84 | 10.61 | 87.63 |
| PC4 | 0.70 | 8.75 | 96.39 |
| PC5 | 0.15 | 1.90 | 98.29 |
| PC6 | 0.09 | 1.14 | 99.43 |
| PC7 | 0.04 | 0.53 | 99.97 |
| PC7 | 0.002 | 0.028 | 100 |

In order to understand the importance of each variable in the construction of the two components, the correlation between the original variables and the components was calculated, and the correlations between the two PCs and their weighting coefficients for each trait are presented in Table 3.

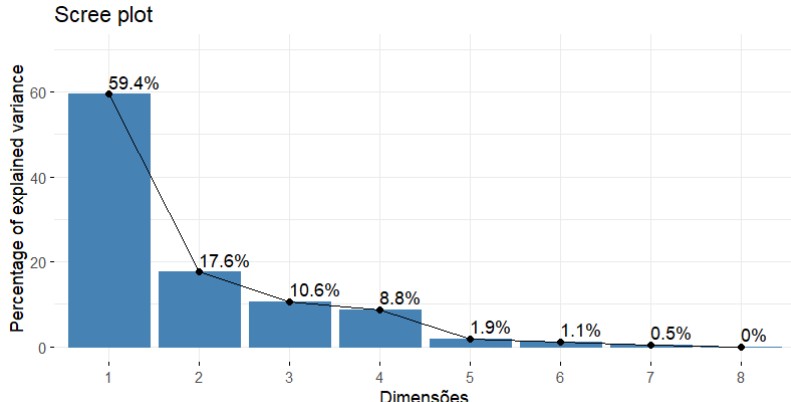

**Figure 11.** The first two PCs explain 77.02% of the variance.

**Table 3.** Weighting coefficients of the characteristics and their correlation with the two PCs.

| Variable | Weighting Coefficient | | Correlation | |
|---|---|---|---|---|
| | PC1 | PC2 | PC1 | PC2 |
| AT | −0.45 | −0.009 | −0.99 | 0.01 |
| RH | 0.44 | 0.12 | 0.97 | −0.15 |
| SH | −0.06 | 0.65 | −0.15 | −0.77 |
| WS | 0.42 | −0.12 | 0.93 | 0.14 |
| SVF | −0.11 | −0.58 | −0.25 | 0.70 |
| Albedo | 0.43 | 0.05 | 0.94 | −0.07 |
| WD | 0.10 | −0.45 | 0.23 | 0.53 |
| LST | −0.44 | 0.02 | −0.97 | −0.03 |

Hougyu et al. [43] evaluated the decision to select two PCs from eight original variables as reasonable; therefore, one can use only the first two PCs for the composition of Equations (2) and (3):

$$PC1 = -0.45\ AT + 0.44\ RH - 0.06\ SH + 0.42\ WS - 0.11\ SVF + 0.43\ Albedo + 0.10\ WD - 0.44\ LST \qquad (2)$$

$$PC2 = -0.009\ AT + 0.12\ RH - 0.65\ SH - 0.12\ WS - 0.58\ SVF + 0.054\ Albedo - 0.45\ WD + 0.02\ LST \qquad (3)$$

According to Equation (2) and Table 3, in PC1, the variables AT, RH, WS, albedo and LST stand out, and the contrast between AT and RH is evident, which can be called a contrast component between air temperature and relative humidity. In Equation (3), the SH, SVF and WD stand out, since the variation explained in PC1 is independent of the variation explained in PC2.

The principal component analysis shows the loading of the sample clusters based on their similarity (Figure 12). The gray dots are the samples, while the lines correspond to the eigenvectors of the principal components. The variables WS and albedo presented similar contributions to PC1 because these variables have vectors of similar length and are closer to the PC1 axis. The contributions of the variables AT and RH are similar, but with opposite signs evidencing the inversely proportional behavior of such variables. The variables on the same side approximate the variance of the value according to their similarity. The points extracted from the Udia clipping are strongly correlated with AT and LST, while the Sintra points are strongly correlated with the WS, albedo and RH variables.

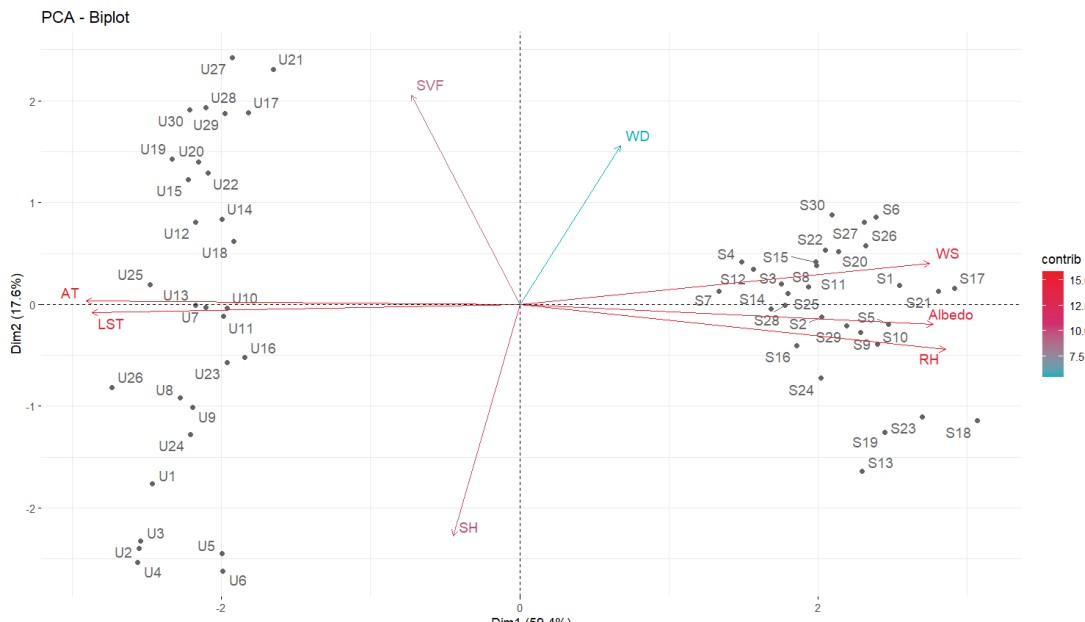

**Figure 12.** Visualization of principal component analysis for two clusters.

Based on this multivariate analysis, it is necessary to analyze the physical composition of each clipping that certainly influences the AT, LST, WS, albedo and RH variables.

### 3.2. Physical Composition and Albedo of the Clippings

When analyzing the physical composition of the clippings and comparing the industrial clippings of Sintra (UI Sin) and Uberlândia (UI Udia), some relevant data can be noticed: the percentages of vegetation and roofs are 2.81% and 8% higher in UI Sintra, respectively, while UI Udia presents 6.52% more impervious surface and 4.17% more exposed soil than UI Sintra, as can be seen in Figure 13.

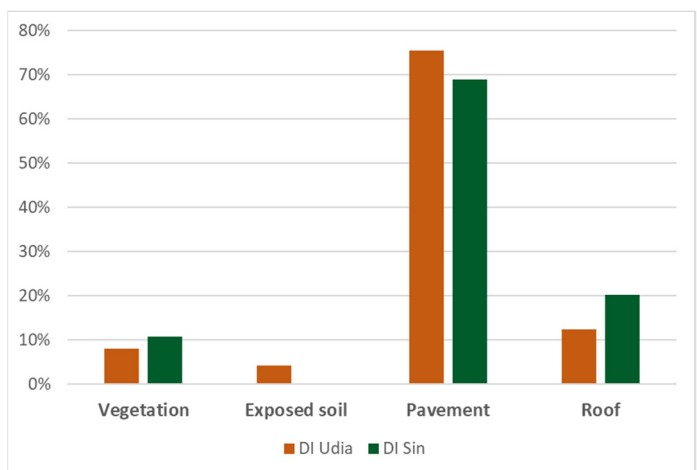

**Figure 13.** Physical composition of the studied cuttings.

A large impermeable space can modify the energy and mass balances, which impairs the local water balance, because it increases surface runoff and decreases evapotranspiration, a reflection of the removal of vegetation for the construction of infrastructure, as pointed out by [2].

The importance of vegetation and water bodies in mitigating the intensity of UHIs is explained by Gunawardena et al. [52]: green spaces dominated by trees tend to offer greater relief from thermal stress and their increased roughness is responsible for cooling

the boundary layer, while "blue spaces" require more planning to yield improvements in the surrounding air temperature.

Such analysis suggests that the influence of evapotranspiration-based cooling from green and blue space is mainly relevant to urban canopy layer conditions, and that tree-dominated green space offers the greatest thermal stress relief when it is most needed. However, the magnitude and transport of the cooling experienced depend on the size, extent and geometry of the green spaces, as some large solitary parks are only capable of offering minimal boundary layer cooling [52].

In addition to evapotranspiration, the shading provided by trees contributes to the reduction in air and surface temperatures, reducing the number of hours of thermal discomfort by up to 21% due to the insertion of green structure and 30% due to vegetation, according to research developed by Palomo Amores et al. [53] in Seville, Spain.

Mello et al. [54] studied the influence of building materials, especially roof tiles, on the urban climate in the interior of Brazil. The samples studied were ceramic, metal and fiber cement. The results showed that fiber cement roofing produced temperatures between 10 °C and 12 °C higher than the surrounding air, ceramic roofing produced temperatures between −0.1 °C and −0.3 °C higher, and metal roofing, used in sheds and services, produced temperatures between 16 °C and 20 °C higher. It was proven that the average temperatures of the city of Presidente Prudente/SP have increased by about 2.5 °C in the past 30 years, with a difference between rural and urban areas of up to 10 °C.

Table 4 gives a brief comparison of the theoretical and measured albedo properties of the sidewalk, roof and vegetation.

**Table 4.** Albedo comparison.

| | Theoretical Albedo | | Measured Albedo | |
|---|---|---|---|---|
| **Category** | **UI Sin** | **UI Udia** | **UI Sin** | **UI Udia** |
| Vegetation | 0.27 * | 0.27 * | 0.28 | 0.21 |
| Roof | 0.57 ** | 0.57 ** | 0.50 | 0.26 |
| Pavement | 0.50 * | 0.20 * | 0.30 | 0.15 |
| Average | 0.44 | 0.34 | 0.36 | 0.20 |
| Standard Deviation | 0.12 | 0.16 | 0.09 | 0.04 |

* Provided by ENVI-met; ** Ferreira [55].

By analyzing the albedo of the surfaces using images from the Landsat 8 satellite and the calculations presented in Equation (1) and shown in Figure 13, it is possible to extract data from this property, presented in Table 4.

At first, it was assumed that the aged albedo of the metal roofs in Sintra and Udia would be the same because they are sandwich roofing sheets. However, the results in Table 4 show that the albedo of the Sintra UI roof is 48% higher than that presented in UI Udia. The average albedo measured in Sintra is, on average, 45% higher than the albedo of Uberlândia for the region studied. This result had the most significant contribution from the albedos of the roofs and sidewalks.

According to Alchapar et al. [56], increasing albedo in the urban environment can improve the thermal conditions of outdoor spaces, mainly by decreasing the maximum temperature, when combined with scenarios with a vegetation percentage above 20%. In addition to the decrease in air temperature during the day, such a change can improve air quality and reduce air conditioning costs and the absorption of solar radiation by surfaces. Using reflective materials on roofs and floors can increase the albedo of each surface by 0.25 and 0.15, respectively, and the total albedo by 0.1 [57]

Roofs play a key role in diminishing the internal temperature of buildings and their surroundings: according to a study by Murguia et al. [58], the albedo of cool roofs can decrease the energy consumption inside buildings, leading to energy savings in commercial

buildings in the USA, due to the action of two mechanisms: solar reflectance and thermal emissivity. Reflectance is the amount of solar energy reflected by the roof and tends to decrease over time. High reflectance saves energy by reflecting incoming solar radiation into space and tends to decrease over time. Maximum reflectance is generally achieved by white roof products, or cool roofs that are apparently dark in visible aspects, but still able to reflect most of the heat and offer more traditional roof colors [58].

For the analysis of WS, AT, LST and RH were chosen to select control points located in open areas, without buildings and with undergrowth vegetation and calculate the difference between them and the 30 random points inside each cutout of Sintra and Udia to enable the comparison of these variables for the two cities studied. The WS, AT and RH data were obtained from the National Institute of Meteorology database for Uberlândia [34] and the Portuguese Institute of the Sea and Atmosphere [33] for Sintra through meteorological stations. Due to the unavailability of data for LST in these databases, the LST will only be compared between the two industrial cuts studied.

*3.3. Descriptive Analysis*

In this section, the results of the most important variables will be presented for Sin UI and Udia UI, determined from the multivariate analysis, namely WS, AT, LST and RH compared to the control points, except for LST.

The Udia and Sintra clippings showed discrepant behavior throughout the day, regarding the difference in WS between the control points and the points of the UIs, presented in Figure 14. The WS at the Sintra UI has a sharp drop of up to $-1.5$ m/s starting at 13 h and extending into the night. In the same period, the WS at the Udia UI remained close to the control points, presenting an average drop of $-1.2$ m/s.

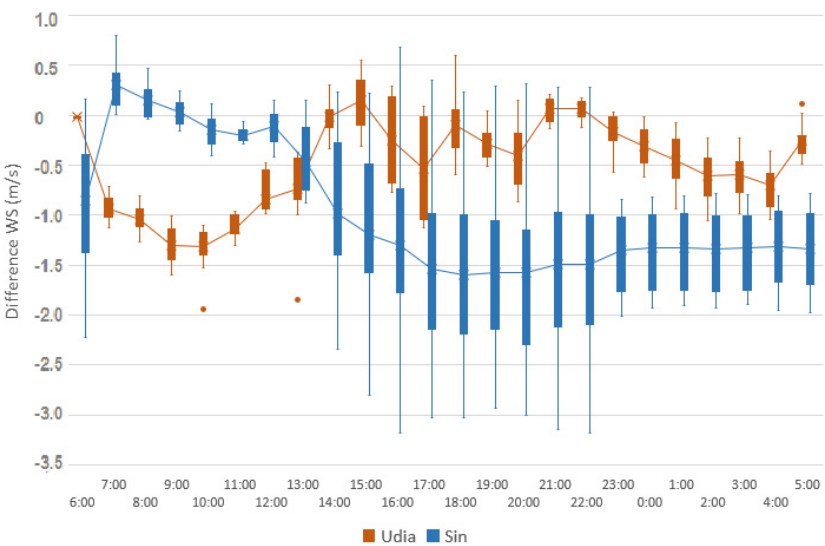

**Figure 14.** Wind speed for UI Sin and Udia on hot days.

Analyzing the RH difference between the points collected for the UIs of Sintra and Udia and control points located in open spaces, it can be seen that there is a greater maximum positive variation for the data from Udia (10%), while in Sintra, there is $-4\%$ variation, meaning that the UI Sin presented the lowest RH. For both cities, the greatest differences in RH occur in the period of heating of the surfaces from 11 to 15 h, as shown in Figure 15.

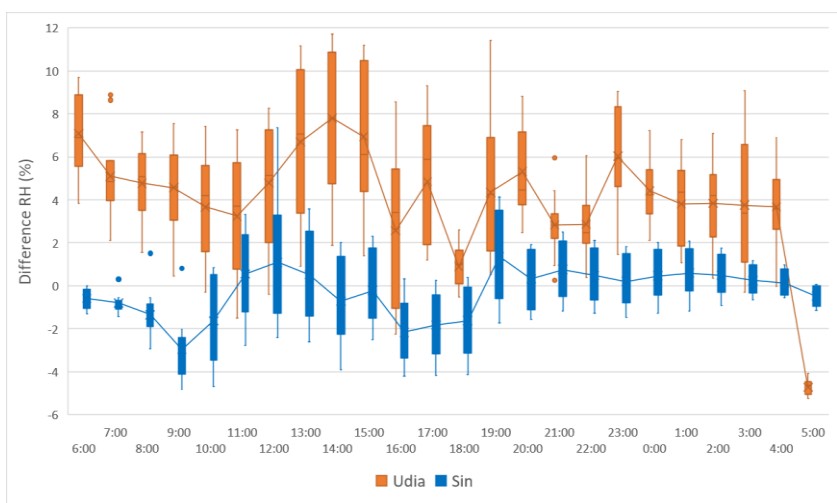

**Figure 15.** Difference in RH of UIs Sin and Udia.

As for the difference in specific humidity, shown in Figure 16, the greatest differences were recorded for the UI Udia (3 g/kg), with all positive values, which indicates that the SH in the UI Udia was greater than that recorded at the points of control. On the other hand, UI Sin did not show considerable SH differences regarding the control points.

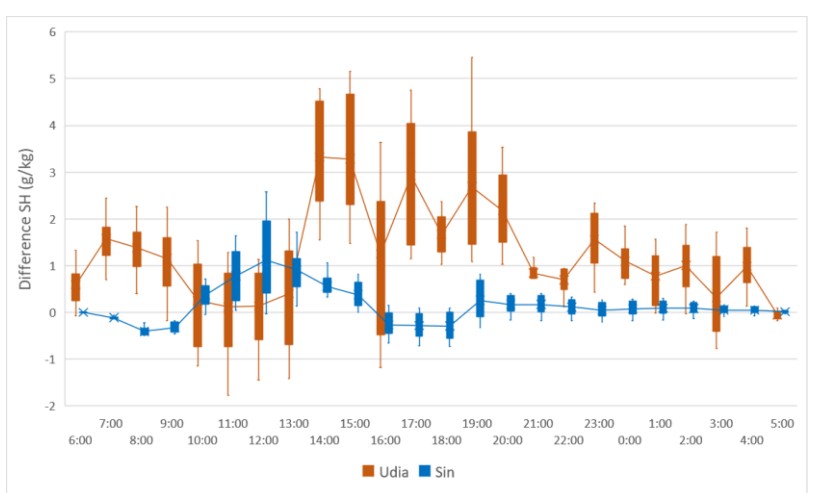

**Figure 16.** Difference in SH of UIs Sin and Udia.

Figure 17 shows the difference in AT of Udia and Sintra and the control points: after sunset, around 19 h, the two clippings show a decrease in the AT difference, remaining approximately constant. For the Sintra UI, the difference between the temperature of the control point and the temperature inside the UI remained positive during almost the entire period, reaching its peak at 12 h, with a difference of 1.2 °C. The Udia clipping, on the other hand, showed an inverse behavior; during the period of radiation incidence, the points inside the UI showed lower air temperatures than the control points, reaching a maximum difference of −1.5 °C at 13 h.

The behavior presented by the points inside the ID Udia can be explained by jointly analyzing the difference in the AT, RH, SH and WS: the period from 13 h to 15 h corresponds to the period with the highest RH variation (8%) and highest SH variation (g/kg), and the inside of the ID is the one with the highest RH compared to the control points, associated with the low WS.

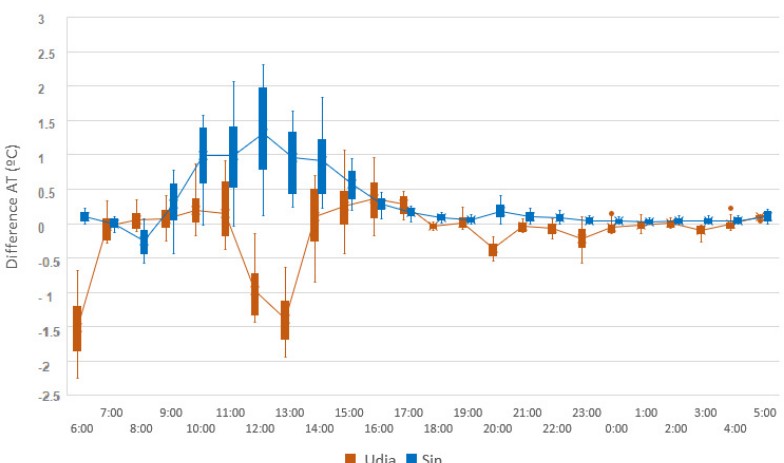

**Figure 17.** Difference in AT of Sin and Udia UIs.

When comparing the LST in the industrial cuts, shown in Figure 18, it is noted that the peak of LST in Sin UI is 25.5% lower than that registered in Udia UI, and the peak hours coincide with the maximum AT regardless of location.

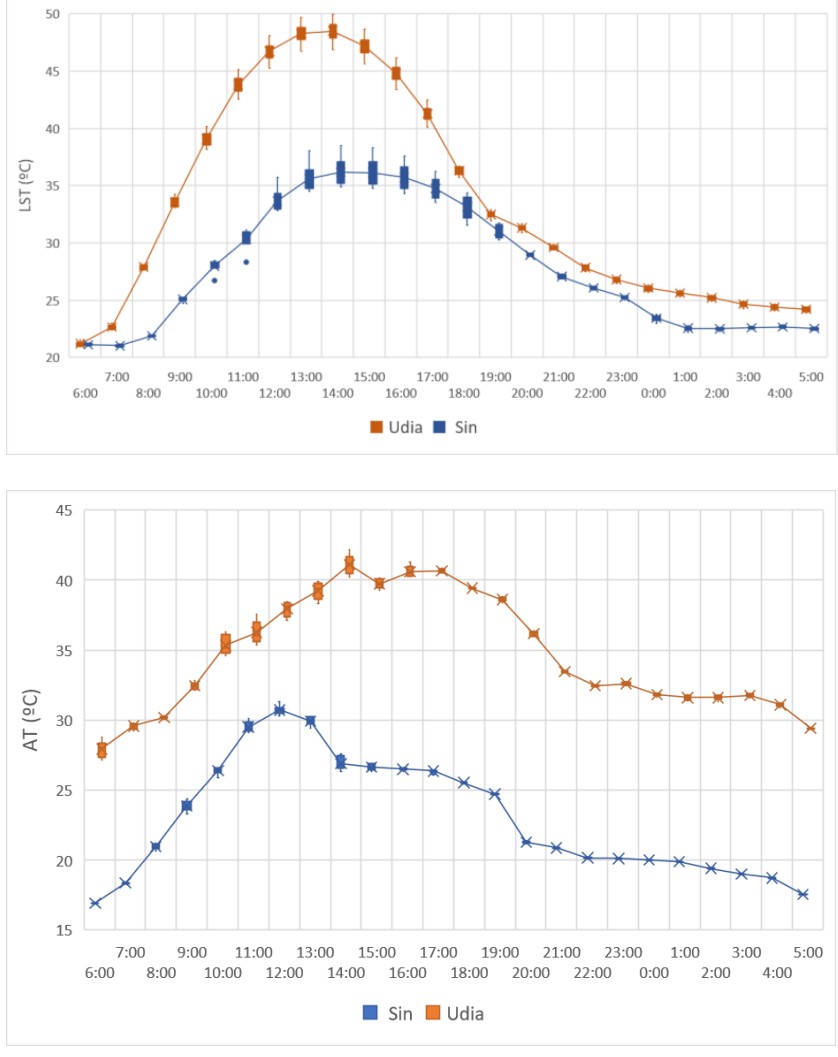

**Figure 18.** Comparison of LST and AT of the Sin and Udia UIs.

## 4. Discussion and Conclusions

This section presents the main topics involved in the study of the microclimate of each scenario studied. From the results obtained, through computer simulation and multivariate analysis, it was possible to determine the existence of heterogeneous groups according to the location, Sintra or Udia. This study allowed us to analyze which among the variables AT, RH, SH, WS, WD, SVF, albedo and LST presented a greater influence on the industrial urban microclimate of Udia and Sintra. For the UI of Udia, the AT and LST were determining factors, while RH, WS and albedo showed a significant influence in Sintra.

Through the characterization of the studied sections, one can perceive the main discrepancies identified between the two sections whose impact on the studied variables will be discussed: the percentage of built area and paved area, and the albedo. UI Udia had a percentage of waterproofed surface 6.52% higher than UI Sin, while the latter had 2.81% more vegetation. The albedo of UI Sin was 45% higher than UI Udia; that is, the building materials present in this sample and their physical composition contributed to the higher reflectance than UI Udia, proving the impact of this variable on the microclimate of UI Sin according to the PCA analysis.

Galusic [59] points out that floors also play an important role in urban air temperature, since surfaces with a high albedo and emissivity remain cooler when exposed to solar radiation, as they absorb less radiation and emit more thermal radiation into space and therefore less heat to the surroundings.

As a comparative parameter, control points were adopted in each city, which are located in regions with low vegetation and no buildings, are fully exposed to solar radiation and lose moisture more easily, and do not benefit from the shading effect. Comparing the RH and SH between the UIs and control points, it was found that both the RH and the SH of the UI Udia were higher than those found at the control points, with peaks of 8% and 3 g/kg, respectively, contrary to UI Sin, whose difference remained close to zero.

The UI Sin and UI Udia presented close values, in modulus, for the difference between AT and the control points during the heating period of the surfaces (9 h to 16 h). UI Sin presented a peak of 1.2 °C higher at the control points at 13 h, and at the same time, UI Udia showed −1.3 °C. Thus, the behavior regarding AT, RH and SH of the studied IUs was different.

In Uberlândia, Silva et al. [46] demonstrated air temperature differences of up to 2.2 °C in winter and 1.2 °C in summer for peripheral neighborhoods with many vacant lots, exposed soil and sparse vegetation.

The LST of the UI Udia was higher than that presented in the UI Sin for the entire simulated period, reaching a peak of 48.49 °C at 15 h, following the AT trend. The LST is mainly influenced by AT because of its control over sensible heat exchange between the Earth's surface and the atmosphere, with heat flowing from the warmer surface to the cooler atmosphere [1].

The positive correlation of LST and AT can be noticed in both clippings as a result of the heat emitted by industries and factories, intense traffic of heavy vehicles and high energy consumption. For urban areas, the higher LST due to the effect of ICUs can cause an increase in AT, creating an area of low pressure that draws in cooler air from surrounding areas [2].

The impermeable surface cover fraction is positively and exponentially correlated with LST, unlike vegetation and water. The impact of impermeable surfaces on regional LST change can be six times greater than the sum of vegetation and water contributions. In percentage terms, Xu, Lin and Tang [60] point out that an addition of 10% of green space or water for each 10% reduction in impermeable surface can decrease the LST by up to 2.9 °C or 2.5 °C, respectively. As for the importance of vegetation, Xiao et al. [61] studied the impact of green areas in industrial districts in China, and the results show that the cooling and humidification effect of medium-sized green spaces was more significant during the hours of high temperatures during the day. In addition, the result shows that the shape

and size of the area within a green space have a significant influence on local cooling and humidification.

The effect of soil sealing causes an increase in air temperature and a decrease in humidity. Thus, preventing the water evaporation process is one of the most relevant aspects to be considered when choosing materials for urban surfaces. The porosity and roughness of buildings, ventilation conditions and the materials of vertical coverings impact in a complementary way the microclimatic conditions of a site [59].

Comparing the difference between the WS at the UIs and the control points, negative values are identified for UI Sin and UI Udia, reaching up to $-1.9$ m/s and $-1.2$ m/s, respectively; that is, the WS in the control points is greater than that recorded in the UIs. The less pronounced difference for the UI Udia may be a consequence of the lower percentage of the built area ($-40\%$) compared to the UI Sintra, since the higher built density provides physical obstacles to the passage of the wind, reducing its speed [2].

The greater Lisbon region benefits from strong winds, considered crucial factors for the decrease in the intensity of UHI at nightfall, acting as a relief factor in the overheating of the region, according to the Report for the Identification of UHIs, because in critical areas, the highest intensity of UHI occurs at dusk, with anomalies > 2 °C, and may exceed 4 °C, decreasing to 1.1 °C–1.5 °C at dusk [37,62]. The role of ventilation in the city of Lisbon was evidenced by Lopes [63], Lopes et al. [64] and Matias and Lopes [65] in their studies of the surface radiation balance of urban materials and its impact on micro-scale air temperature in a Lisbon neighborhood. The authors concluded that both temperatures and the radiative balance of facades and surfaces respond directly to the incident solar radiation and that when streets are not benefited by the prevailing wind direction, air temperatures tend to be higher compared to streets exposed to the wind. However, in Sintra, a lower WS at the UI Sin compared to the control points had a negative impact on the AT reduction, especially in the afternoon.

Therefore, comparing the Sintra and Udia UIs, it can be concluded that even though the Udia UI has materials with lower albedo, which demand more time to heat up and heat loss occurs more slowly after sunset, and it also has lower percentages of vegetation, relative humidity and wind speed than the Sin UI, the air temperatures inside it may be lower than in the unshaded surroundings. Although the microclimate of the Sin UI is dependent on other variables, the WS, whose function is the cooling and loss of heat from the surface to the environment [2], had a significant impact and a difference of $-1.9$ m/s compared to the control points; this parameter caused a peak of 1.5 °C in the industrial environment at 13 h. Therefore, urban ventilation planning can contribute to sustainable urban development, considering primary ventilation corridors where the wind direction is predominant, parks and green spaces, and complementary secondary ventilation corridors [66].

This study is a useful tool to identify the most important variables for the industrial microclimate of two different cities. In future research, we intend to carry out measurements in situ to verify if the conclusions obtained through the simulations are confirmed and to expand the study to other types of LCZs. Since the albedo of the Sintra roofs and sidewalks was responsible for raising the albedo of the cutout as a whole, it is proposed to increase the albedo of Udia roofs by applying acrylic paint with high reflectivity and installing sidewalks with higher albedo. Such changes are expected to have a positive impact on decreasing the AT in the Udia UI, as in the Sintra UI, although it lacks demonstration.

**Author Contributions:** Conceptualization, A.M.d.A., A.L. and É.M.; methodology, A.M.d.A.; formal analysis, A.M.d.A.; writing—original draft preparation, A.M.d.A., A.L. and É.M.; writing—review and editing, A.M.d.A., A.L. and É.M.; supervision, A.L. and É.M. All authors have read and agreed to the published version of the manuscript.

**Funding:** This study was financed in part by the Coordenação de Aperfeiçoamento de Pessoal de Nível Superior—Brasil (CAPES)—Finance Code 001, and FCT—Fundação para a Ciência e Tecnologia, I.P. (CEG projects numbers: UIDB/00295/2020 and UIDP/00295/2020).

**Conflicts of Interest:** The authors declare no conflict of interest.

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
