# Peer review of "Microclimate Multivariate Analysis of Two Industrial Areas"

_atmosphere, doi:10.3390/atmos14081321_

Round 1

Reviewer 1 Report

The study selects multiple influencing factors such as "air temperature, relative and specific humidity, wind speed and direction, sky view factor and the albedo of the material surfaces" to analyze their impact on the urban microclimate. The research has certain novelty and can be published in this journal. However, before that, there are many issues that need to be revised.

(1) The most direct manifestation of the change of urban microclimate effect is the change of urban land surface temperature. So why remove the effect of land surface temperature in the study?

(2) The introduction is too scattered. Some related but short paragraphs can be merged.

(3) In Figure 2, Figure 3, the latitude and longitude gap are too small, it is recommended to expand to degrees and minutes.

(4) Figure 5 should have a legend; We cannot tell what color corresponds to what type of LCZ. Also, is "(b) UI Udia" classified correctly? It seems to be far from "b" in Figure 4. It is recommended to increase the number of training samples when classifying, or use drones to collect building data for LCZ classification.

(5) The references appear to be in the wrong place of citation.

(6) Figure 8 should contain a legend.

(7) L282-L285 These paragraphs could be merged. L348-L362 These paragraphs could be merged.

(8) The text part still needs to be condensed, and the structure of the article also needs to be further optimized. Are the Results and Discussion considered to be written separately?

The study selects multiple influencing factors such as "air temperature, relative and specific humidity, wind speed and direction, sky view factor and the albedo of the material surfaces" to analyze their impact on the urban microclimate. The research has certain novelty and can be published in this journal. However, before that, there are many issues that need to be revised.

(1) The most direct manifestation of the change of urban microclimate effect is the change of urban land surface temperature. So why remove the effect of land surface temperature in the study?

(2) The introduction is too scattered. Some related but short paragraphs can be merged.

(3) In Figure 2, Figure 3, the latitude and longitude gap are too small, it is recommended to expand to degrees and minutes.

(4) Figure 5 should have a legend; We cannot tell what color corresponds to what type of LCZ. Also, is "(b) UI Udia" classified correctly? It seems to be far from "b" in Figure 4. It is recommended to increase the number of training samples when classifying, or use drones to collect building data for LCZ classification.

(5) The references appear to be in the wrong place of citation.

(6) Figure 8 should contain a legend.

(7) L282-L285 These paragraphs could be merged. L348-L362 These paragraphs could be merged.

(8) The text part still needs to be condensed, and the structure of the article also needs to be further optimized. Are the Results and Discussion considered to be written separately?

Reviewer 2 Report

Hard to understand what the authors will do.

Reviewer 3 Report

This study aims to analyze, in addition to land surface temperature other thermal variables such as: air temperature, relative and specific humidity, wind speed and direction, sky visibility factor and albedo of material surfaces. With the intention of verifying which of these has the greatest impact on urban microclimate.

Please see below further comments for consideration:

In the Abstract, the reviewer suggests adding numbers that can support the results obtained from the study, which were in any case expressed in words to lines 20-23.

In the initial part of the introduction, at lines 44-50, the reviewer suggests including sentences describing the different strategies that the scientific literature proposes to counteract the development of UHIs. Therefore, I propose to study and refer on the following recent studies:

https://doi.org/10.3390/su142316027

https://doi.org/10.3390/atmos14050857

The reviewer suggests moving lines 122-137 to the Methods section because they describe the case study of the article.

Improve the quality of the resolution of Figure 2.

The reviewer suggests calling air temperature by AT instead of Tair because it is more popular in the scientific literature.

Part of the results, such as lines 492-510 could make up an interesting section of the Discussion.

Finally, in the Conclusions section, the reviewer suggests adding numbers to the results and describing potential future applications of the proposed study.

The quality of writing in this manuscript is sufficient, so moderate editing of the adopted English is recommended.

Round 2

Reviewer 1 Report

The authors made changes in accordance with the reviewer's recommendations. The  manuscript can be published without re-reviewing.

Reviewer 3 Report

The authors followed the proposed suggestions excellently.